# Splenic Hilar Involvement and Sinistral Portal Hypertension in Unresectable Pancreatic Tail Cancer

**DOI:** 10.3390/cancers15245862

**Published:** 2023-12-15

**Authors:** Takeshi Okamoto, Tsuyoshi Takeda, Takafumi Mie, Tatsuki Hirai, Takahiro Ishitsuka, Manabu Yamada, Hiroki Nakagawa, Takaaki Furukawa, Akiyoshi Kasuga, Takashi Sasaki, Masato Ozaka, Naoki Sasahira

**Affiliations:** Department of Hepato-Biliary-Pancreatic Medicine, Cancer Institute Hospital of Japanese Foundation for Cancer Research, 3-8-31 Ariake, Koto-ku, Tokyo 135-8550, Japan; tsuyoshi.takeda@jfcr.or.jp (T.T.); takafumi.mie@jfcr.or.jp (T.M.); tatsuki.hirai@jfcr.or.jp (T.H.); takahiro.ishitsuka@jfcr.or.jp (T.I.); manabu.yamada@jfcr.or.jp (M.Y.); hiroki.nakagawa@jfcr.or.jp (H.N.); takaaki.furukawa@jfcr.or.jp (T.F.); akiyoshi.kasuga@jfcr.or.jp (A.K.); takashi.sasaki@jfcr.or.jp (T.S.); masato.ozaka@jfcr.or.jp (M.O.); naoki.sasahira@jfcr.or.jp (N.S.)

**Keywords:** pancreatic cancer, spleen, left-sided portal hypertension, gastric varices, upper gastrointestinal bleeding

## Abstract

**Simple Summary:**

Pancreatic cancer can be categorized into pancreatic head, body, and tail cancer, all of which carry a grim prognosis. Pancreatic tail cancer frequently invades the splenic hilum and splenic vein. It is currently unclear whether splenic hilar involvement is associated with a poor prognosis. Splenic vein occlusion can also lead to sinistral portal hypertension, which can in turn give rise to gastric varices. The clinical impact of sinistral portal hypertension and hemorrhagic events has yet to be elucidated. In this study, we explore the effect of splenic hilar involvement and sinistral portal hypertension on outcomes in patients with pancreatic tail cancer. The results of this study may contribute to better prediction of outcomes and assist physicians in the education and care of their patients.

**Abstract:**

Background: Pancreatic tail cancer (PTC) frequently displays splenic hilar involvement (SHI), but its impact on clinical outcomes remains unclear. We investigated the clinical impact of SHI in patients with unresectable PTC. Methods: We retrospectively reviewed all patients with unresectable PTC who received first-line therapy at our institution from 2016 to 2020. Results: Of the 111 included patients, 48 had SHI at diagnosis. SHI was significantly associated with younger age, liver metastasis, peritoneal dissemination, larger tumor size, modified Glasgow prognostic score of 1 or more, splenic artery involvement, gastric varices, and splenomegaly. Shorter median overall survival (OS; 9.3 vs. 11.6 months, *p* = 0.003) and progression-free survival (PFS; 4.3 vs. 6.3 months, *p* = 0.013) were observed in SHI patients. Poor performance status of 1 or 2, tumor size > 50 mm, hepatic metastasis, mGPS of 1 or 2, and SHI (hazard ratio: 1.65, 95% confidence interval: 1.08–2.52, *p* = 0.020) were independent predictors of shorter OS. Splenic artery pseudoaneurysm rupture and variceal rupture were rare and only observed in cases with SHI. Conclusions: Splenic hilar involvement is associated with worse outcomes in pancreatic tail cancer.

## 1. Introduction

Differences in characteristics of pancreatic ductal adenocarcinoma (PDAC) based on its origin (dorsal or ventral pancreas) and location (pancreatic head, body, or tail) have been investigated in various studies [1,2,3,4,5,6]. A majority of PDACs occur in the pancreatic head, which is more prone to fatty tissue infiltration and has fewer islets of Langerhans than the body or tail [3]. PDACs of the pancreatic head are also more likely to be detected early based on symptoms such as obstructive jaundice, with reports suggesting longer overall survival (OS) compared to PDACs of the body or tail [3,4] but shorter survival when matching cancer stage [5,6,7,8]. Recently, differences in genomic composition based on PDAC location have been found, with more enriched genomic alterations but lower rates of druggable targets in PDACs of the body or tail [9,10]. While PDACs of the pancreatic body and tail are often grouped together in comparison with pancreatic head cancers, behavioral differences between PDACs arising in the pancreatic body and those arising in the pancreatic tail have also been reported [11].

The pancreatic head and body are anatomically close to key arteries such as the celiac axis, superior mesenteric artery, and their respective branches, as well as the splenoportal confluence (SPC). On the other hand, PDAC of the pancreatic tail (pancreatic tail cancer: PTC) initially only invades the splenic vein and/or artery, making them less likely to be borderline resectable [12] or unresectable due to locally-advanced disease. However, splenic vein occlusion can give rise to sinistral portal hypertension (SPH) and complications such as gastric varices (GV) [13]. SPH has been reported after pancreaticoduodenectomy involving splenic vein or SPC resection [14,15,16,17,18], but reports on SPH in unresectable PDAC are scarce [13].

A unique feature of PTC is its ability to invade the splenic hilum. There are sparse reports on splenic hilar involvement (SHI) in PTC from surgeons, evaluating the techniques and potential benefits and harms of spleen-preserving distal pancreatectomy [19,20,21]. However, the clinical implications of SHI in unresectable PTC has not been studied. Therefore, we investigated the radiological findings and outcomes of patients with unresectable PTC, focusing on the presence of SHI and SPH.

## 2. Materials and Methods

### 2.1. Patients

We reviewed records of all patients with unresectable PDAC of the pancreatic tail who underwent chemotherapy at our institution between 1 January 2016 and 31 December 2020, from our prospectively maintained database. Only patients with pathologically-proven PTC that was unresectable at the time of cancer diagnosis and with at least one contrast-enhanced computed tomography (CT) examination at our institution were included. Exclusion criteria included co-existing advanced-stage cancers, history of gastrectomy or splenectomy, extension of PDAC to the pancreatic head or body, and malignant SPC obstruction. Cases of isolated portal vein thrombosis were included if determined not to be tumor thrombi based on the lack of contrast enhancement on contrast-enhanced CT.

### 2.2. Baseline Characteristics

Age, Eastern Cooperative Oncology Group performance status (PS), disease status, sites of metastases, and laboratory data including tumor markers were evaluated at the time of diagnosis. Modified Glasgow prognostic score (mGPS) was calculated based on serum albumin and C-reactive protein (CRP) at diagnosis, with a score of 0 if CRP is ≤1 mg/dL, a score of 1 if albumin is ≥3.5 g/dL and CRP is ˃1 mg/dL, and a score of 2 if albumin is ˂3.5 g/dL and CRP ˃ 1 mg/dL [22]. Neutrophil-to-lymphocyte ratio (NLR) was calculated as the ratio of neutrophils to lymphocytes in the white blood cell differential [23].

Contrast-enhanced CT was performed within one month of diagnosis in all patients, and such findings were assumed to reflect the baseline condition at diagnosis (i.e., deemed to be taken on the day of diagnosis). Axial images were captured in 1.25 mm slices and were reconstructed for coronal images. In general, CT images were evaluated by board-certified radiologists specializing in oncology, and measurements were obtained by one of the authors (TO) using both axial images and coronal reconstruction.

Tumor size was measured along the splenic vein to reflect the degree of splenic vein invasion, which generally, but not always, was also the maximum diameter of the tumor. Vessel invasion was considered present if there was any degree of stenosis in the target vessel, while occlusion was considered present if no contrast passed through the invaded portion of the target vessel. SHI was considered present if the tumor invaded the splenic hilum causing stricture or occlusion where vessels branched out immediately before entering the spleen (Figure 1). Direct contact with, or invasion into, the spleen was therefore not required to meet the definition of SHI. Dilation of the short, posterior, or left gastric veins was defined as a maximum diameter of 5 mm or more at any point along the relevant vein. Splenic index was calculated as largest measurement in the axial plane x largest perpendicular dimension to such measurement x maximum craniocaudal length on coronal reconstruction, and splenomegaly was defined as having a splenic index > 480 [24,25]. The presence of varices was determined based on collateral veins reaching the esophageal or gastric mucosa on contrast-enhanced CT, irrespective of whether varices could be identified on esophagogastroduodenoscopy (EGD). Varices were classified according to Sarin et al.’s classification [26] and were used as a surrogate marker for SPH in this study.

EGD was only performed when upper gastrointestinal bleeding (UGIB) was suspected due to hematemesis, coffee ground emesis, melena, or a sudden drop in hemoglobin; no routine screening EGDs were performed. The cause of UGIB was determined based on EGD reports and images. All procedures were performed by board-certified endoscopists or trainees under their direct supervision. Endoscopic hemostasis was attempted at the discretion of the endoscopist.

### 2.3. Follow-Up

Contrast-enhanced CT was performed every 2–3 months except in cases that developed kidney injury or allergies to contrast media during the follow-up period, in which case CT without contrast was performed. The maximum diameter of varices was determined by measuring the largest varix (as defined above) at the esophageal or gastric mucosa found in all contrast-enhanced CT scans performed on each patient. The measurement was generally based on the last contrast-enhanced CT performed, unless factors such as massive ascites precluded an accurate measurement.

Given the limited prognosis associated with unresectable PDAC, patients were not scheduled for follow-up endoscopy, except for a second-look endoscopy in one patient.

Chemotherapy regimens were selected at the discretion of the oncologist. Treatment was continued until disease progression, patient refusal, intolerable toxicity, inability to continue treatment for other reasons, or conversion surgery. Response to chemotherapy was evaluated in accordance with the response evaluation criteria in solid tumors (RECIST) guideline (version 1.1) [27]. OS was defined as the time from the starting date of first-line therapy until death from any cause or the last follow-up date. Progression-free survival (PFS) was defined as the time from the starting date of first-line therapy until death from any cause, disease progression confirmed on imaging studies, or the last follow-up date. Follow-up data were confirmed up to 30 April 2023.

### 2.4. Statistical Analysis

Categorical variables are shown as absolute numbers (percentages), while continuous variables are shown as medians (interquartile range). Denominators of ratios were adjusted for missing data. Statistical analyses were conducted using Pearson’s chi-square test or Fisher’s exact test, as appropriate, for categorical variables and the Mann–Whitney U test for continuous variables. Survival analysis using Kaplan–Meier curves and log-rank analysis was conducted to evaluate OS and PFS. Cox regression analysis was performed to evaluate factors affecting OS. Given the limited sample size, multivariate analysis was performed on five variables that were considered most significant in univariate analysis. Spearman’s correlation coefficient was calculated to evaluate correlation between variables. The *p* values < 0.05 were considered statistically significant. Statistical analyses were performed using IBM SPSS Statistics ver. 28.0 (IBM Corp., Armonk, NY, USA).

### 2.5. Ethical Considerations

The institutional review board at our hospital gave approval for this study (2023-GB-033). Patient consent was waived due to its retrospective nature. The study was publicized on the hospital website, giving patients the opportunity to opt out of the study without any impact on their care.

## 3. Results

### 3.1. Patient Characteristics

Of the 651 patients who started chemotherapy for unresectable PDAC during the study period, a total of 111 patients (17.1%) met the inclusion criteria, of which 48 had SHI (7.4%) at diagnosis (Table 1). Patients with SHI were significantly younger than those without (median age: 64 vs. 69 years, *p* = 0.011). The SHI group was also more likely to have liver metastasis (79.2% vs. 57.1%, *p* = 0.015), peritoneal dissemination (51.2% vs. 33.3%, *p* = 0.047), higher mGPS (52.1% vs. 71.4% with mGPS of 0, *p* = 0.036), larger median tumor size (53 vs. 42 mm, *p* < 0.001), splenic artery occlusion (43.8% vs. 7.9%, *p* < 0.001), gastric varices on contrast-enhanced CT (60.4% vs. 39.7%, *p* = 0.030), and splenomegaly (12.5% vs. 0%, *p* = 0.004) than the non-SHI group, among other factors.

### 3.2. Follow-Up

Follow-up contrast-enhanced CT was conducted on 87.5% and 85.7% (*p* = 0.785) of SHI and non-SHI patients, respectively, for a median period of 7.4 and 8.3 months (*p* = 0.098), respectively (Table 2). GVs ultimately arose in 87.5% of the SHI group and 71.4% of the non-SHI group (*p* = 0.042). The breakdown of GV types were similar between groups. SHI patients were more likely to have GVs drained via the posterior gastric vein than the non-SHI group (23.8% vs. 4.4%, *p* = 0.020).

Emergency EGD was performed for suspected upper gastrointestinal bleeding in seven patients, of which sources of bleeding were identified in six (85.7%) and three (42.9%) showed signs of GV (one patient each had F1, F2, and F3 varices, respectively). Variceal rupture was only observed in one case belonging to the SHI group (*p* = 0.432) (Figure 2). All three cases of splenic artery pseudoaneurysm rupture, of which one was fatal, also belonged to the SHI group (*p* = 0.078).

There were 54 patients with GVs at cancer diagnosis. The characteristics of patients with and without GVs at diagnosis are presented in Table 3 and Table 4. Patients with GVs at diagnosis were more likely to have SHI, be younger, have peritoneal dissemination, have larger tumors, and have larger spleens, but were less likely to have an unaffected pancreatic tail remaining. Thirty-three of the fifty-seven patients without GVs at diagnosis (57.9%) developed GVs during follow-up, of which 94.3% were drained mainly by the left gastric vein.

### 3.3. Chemotherapy and Survival

Chemotherapy regimens and responses to treatment are summarized in Table 5. More patients in the non-SHI group received gemcitabine and nab-paclitaxel combination therapy, although the difference was not significant (*p* = 0.055). While there was no difference in overall response, the SHI group had a lower disease control rate than the non-SHI group (58.3% vs. 77.8%, *p* = 0.006). Shorter median OS (9.3 vs. 11.6 months, *p* = 0.003) and PFS (4.3 vs. 6.3 months, *p* = 0.013) were observed in the SHI group (Figure 3).

Univariate Cox analyses revealed the following significant factors predicting shorter OS: poor performance status (1 or 2), tumor size > 50 mm, hepatic metastasis, high NLR, absence of thrombocytopenia, mGPS of 1 or 2, CA19-9 > 500 U/mL, and SHI (Table 6). Of these, poor performance status (hazard ratio (HR): 1.86, 95% confidence interval (CI): 1.19–2.89, *p* = 0.006), hepatic metastasis (HR: 1.93, 95% CI: 1.22–3.03, *p* = 0.005), mGPS of 1 or 2 (HR: 1.96, 95% CI: 1.25–3.08, *p* = 0.003), and SHI (HR: 1.65, 95% CI: 1.08–2.52, *p* = 0.020) were independent predictors of OS.

With respect to PFS, significant factors in univariate analysis were old age (70 years or older), poor performance status, tumor size > 50 mm, NLR > 4, absence of thrombocytopenia, mGPS of 1 or 2, and SHI (Table 7). Poor performance status, tumor size > 50 mm, absence of thrombocytopenia, and mGPS of 1 or 2 were independent predictors of shorter PFS. SHI fell slightly short of significance (HR: 1.49, 95% CI: 0.98–2.28, *p* = 0.065) in the multivariate analysis. 

Correlation analysis revealed mild positive correlations between SHI and three factors that were significant in univariate Cox analysis: tumor size > 50 mm (rho: 0.274, *p* = 0.004), hepatic metastasis (rho: 0.231, *p* = 0.015), and mGPS of 1 or 2 (rho: 0.199, *p* = 0.037).

The presence of GVs at diagnosis had no significant impact on response to chemotherapy (Table 8) or on survival outcomes (Figure 4).

## 4. Discussion

In this study, we evaluated the impact of SHI and SPH on clinical outcomes in unresectable PTC. SHI at diagnosis was significantly associated with younger age and more advanced disease, including liver metastasis, peritoneal dissemination, larger tumor size, a modified Glasgow prognostic score of 1 or more, splenic artery involvement and occlusion, GVs, and splenomegaly. Shorter median OS and PFS were observed in SHI patients. SHI was significantly associated with OS and PFS in univariate analysis, and with OS in multivariate analysis. Almost 90% of GVs resulting from SPH due to PTC were isolated GVs (IGV). Variceal rupture was rare (0.9%) despite evidence of GV on CT at diagnosis in 48.6% and ultimately in 78.6% of patients. Splenic artery pseudoaneurysm rupture and variceal rupture were only observed in cases with SHI.

Other than the fact that it precludes spleen preserving distal pancreatectomy, the clinical implications of SHI have not been studied in depth. As expected, it was associated with more advanced disease, particularly with tumor size. Kaplan–Meier curves showed that the SHI group had significantly shorter OS and PFS relative to the non-SHI group. SHI was an independent predictor of poor OS, but fell slightly short of significance as an independent predictor of PFS in multivariate analyses, most likely due to its mild but significant correlation with tumor size. We hypothesize that SHI is an early sign of extra-pancreatic tumor invasion and occult peritoneal dissemination. Invasion of all branches of the splenic artery and vein at the splenic hilum may also predispose to metastatic spread to the liver and elsewhere, through the short gastric vein and other collateral vessels.

Direct PTC invasion to nearby structures can lead to duodenal obstruction near the ligament of Treitz (observed in one patient), colonic obstruction near the splenic flexure (observed in seven patients), and most commonly, SPH. SPH is a major cause of GVs, which have a lower bleeding risk than their esophageal counterparts but can bleed severely, leading to mortality in 45% of cases [26]. In general, about 75% of GVs are GOV (gastroesophageal varices) type 1, about 20% are GOV type 2, and the remainder are IGV type 1 or 2. We found the reverse trend in PTC, with IGVs accounting for 89.7% of all GVs. SHI did not affect the location of varices. Patients with SHI were more likely to have patent posterior gastric veins and to have varices drained by them, probably because their tumors originated closer to the spleen. Although the sole case of GV rupture had SHI, the risk of rupture in patients without SHI may not necessarily be low, as there was also a case of F3 GV confirmed on EGD in the non-SHI group.

The rate of splenomegaly was relatively low in this study, being observed in 5.4% of patients at diagnosis and ultimately in 16.2% of patients. The initial splenic index was larger in the SHI group, but the difference fell slightly short of significance (*p* = 0.051). We investigated low platelet count as a surrogate for hypersplenism, hypothesized to reflect greater SPH and therefore a worse prognosis. However, low platelet count was associated with favorable OS (HR: 0.34, 95% CI: 0.12–0.92, *p* = 0.038) and PFS (HR: 0.29, 95% CI: 0.11–0.81, *p* = 0.018) in univariate Cox analysis, and was an independent predictor of PFS but not OS. Because thrombocytosis is associated with tumor progression, metastasis, and a poor prognosis in unresectable pancreatic cancer [28,29], we speculate that the benefits afforded by the absence of thrombocytosis more than offset the disadvantages associated with hypersplenism in this population.

Bleeding was reported in 1.6% of PDAC patients in a large French retrospective study, of which 75% were pancreatic head cancers [30]. The authors found that variceal rupture is the second most common cause of UGIB in pancreatic cancer patients, after gastrointestinal tumor invasion, and that median OS after PDAC-associated UGIB was 2.7 months. In our study, sources of UGIB were identified in six cases, accounting for 0.9% of all unresectable PDAC cases and 5.4% of all unresectable PTC cases. Median OS after UGIB was longer than the previous report, at 7.0 (range: 1.5–28.4) months. In our study, GV rupture occurred only in 0.9% of unresectable PTC cases and in 2.1% of SHI cases. While endoscopic variceal treatment is generally not very effective in SPH due to a myriad of dilated vessels involved [31], our sole patient with GV rupture survived 7.2 months after the UGIB event. We speculate that most unresectable PTC patients do not live long enough to experience variceal rupture. Given the costs and patient burden involved, we believe it is premature to recommend routine EGD screening in all unresectable PTC patients, including those with SHI.

Splenic artery pseudoaneurysm rupture is another potentially fatal complication, generally associated more with pancreatitis than PDAC. Splenic artery pseudoaneurysm rupture due to PDAC is rare, with only six case reports to date, when excluding iatrogenic and post-surgical cases [32,33,34,35,36,37]. Three were pancreatic body cancers and three were PTCs, of which two had SHI, while all cases were treated successfully with transcatheter arterial embolization (TAE). We experienced three PTC cases with splenic artery pseudoaneurysm rupture, all associated with SHI. This complication may therefore be more common than once believed. All cases were treated successfully with TAE, with no notable impact on OS. As SHI was associated with a higher rate of splenic artery occlusion in our study, SHI may increase the risk of splenic artery pseudoaneurysm rupture. However, as cases of pseudoaneurysm rupture without SHI have been reported, more research is required to clarify whether SHI is a risk factor for pseudoaneurysm formation and rupture in PTC patients.

The association between chemotherapy and SHI has not been studied. We found that SHI was associated with a lower disease control rate, with progressive disease observed at the first follow-up CT in 37.5% of patients. Again, SHI may be a sign of more aggressive disease, demonstrating a proclivity towards extra-pancreatic tumor invasion and peritoneal dissemination. While the unfavorable impact on PFS did not remain significant in multivariate Cox analysis, it may be necessary to pay particular attention for rapid disease progression in SHI patients.

This study has several limitations. It was a retrospective study at a single institution with a limited sample size. Reports on SHI are scarce and there is no consensus to date on the definition of SHI, making interpretations of our results difficult. Only seven cases underwent EGD, precluding meaningful comparisons with CT findings. Risk factors for UGIB in unresectable PTC could not be evaluated due to the small number of patients presenting with UGIB. Due to the small sample size, not all significant variables from univariate Cox analysis could be included in multivariate analysis.

## 5. Conclusions

Splenic hilar involvement was significantly associated with younger age, more advanced disease, and shorter survival. As all cases of variceal rupture and splenic artery pseudoaneurysm rupture occurred in patients with splenic hilar involvement, a higher degree of caution for bleeding events may be warranted. Further studies with larger sample sizes may elucidate risk factors for bleeding events in unresectable pancreatic tail cancer.

## Figures and Tables

**Figure 1 cancers-15-05862-f001:**
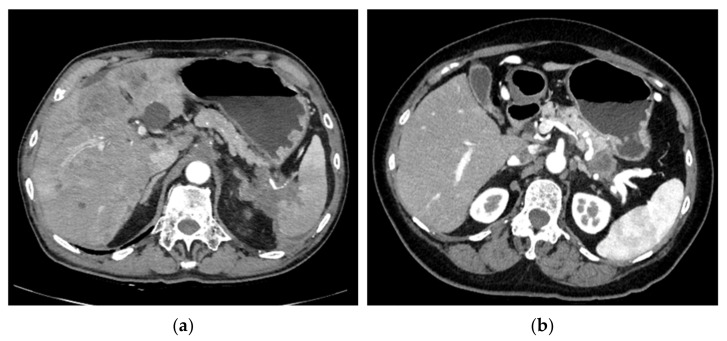
(**a**) A case with splenic hilar involvement. (**b**) A case with no splenic hilar involvement.

**Figure 2 cancers-15-05862-f002:**
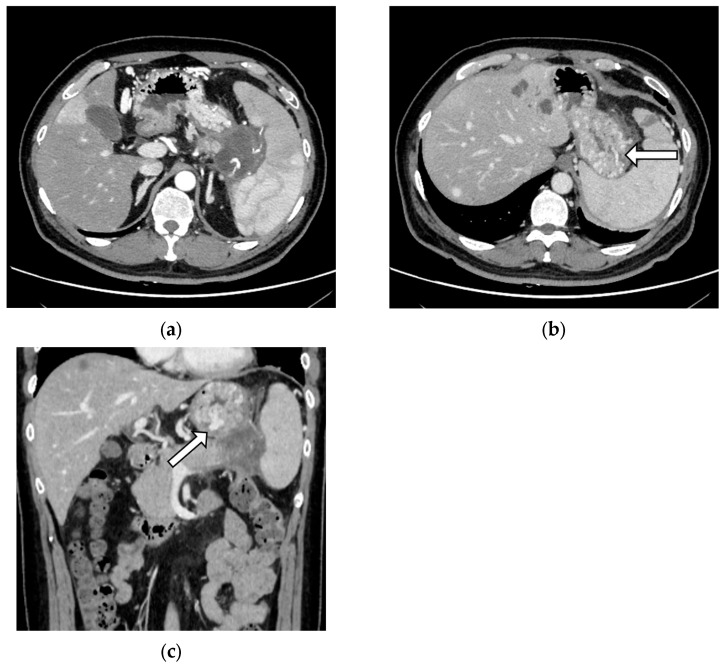
A case of gastric variceal rupture. A 39-year-old man with known liver, lung, and bone metastases of pancreatic tail cancer presented with hematemesis. The most recent computed tomography revealed (**a**) splenic hilar involvement and multiple dilated vessels reaching the surface of the gastric mucosa (arrows) on (**b**) axial and (**c**) coronal images. The patient was successfully treated with endoscopic injection sclerotherapy followed by partial splenic embolectomy at a tertiary referral center.

**Figure 3 cancers-15-05862-f003:**
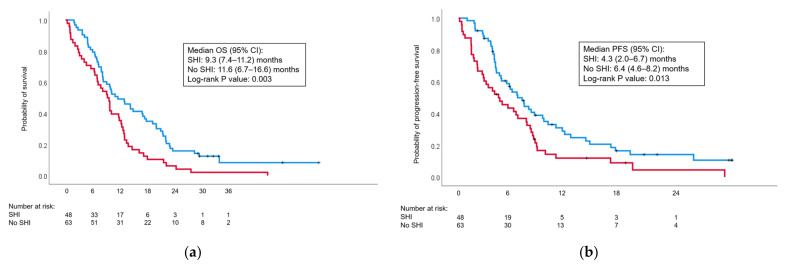
Kaplan–Meier curves comparing (**a**) overall survival (OS) and (**b**) progression-free survival (PFS) in patients with and without splenic hilar involvement (SHI). SHI: red curve; no SHI: blue curve. CI: confidence interval.

**Figure 4 cancers-15-05862-f004:**
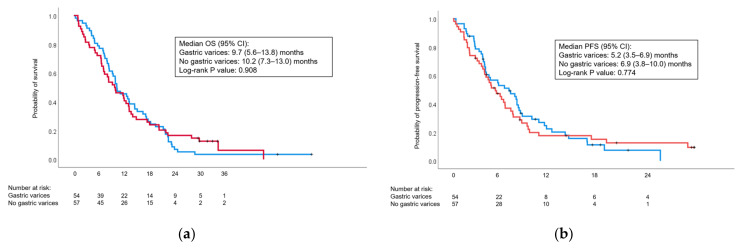
Kaplan–Meier curves comparing (**a**) overall survival (OS) and (**b**) progression-free survival (PFS) in patients with and without gastric varices at diagnosis. Gastric varices: red curve; no gastric varices: blue curve. CI: confidence interval.

**Table 1 cancers-15-05862-t001:** Baseline characteristics.

			Splenic Hilar Involvement				
	Total		Yes		No		
	(*n* = 111)		(*n* = 48)		(*n* = 63)		*p* Value
Age at diagnosis, years, median (IQR)	66	(58–72)	64	(56–70)	69	(62–74)	0.011
Male, %	64	57.7%	29	60.4%	35	55.6%	0.608
Body mass index, median (IQR)	21.8	(19.4–24.1)	22.4	(19.6–24.4)	21.3	(19.2–23.9)	0.271
Performance status, %							
0	75	67.6%	28	58.3%	47	74.6%	0.070
1/2	35/1	32.4%	19/1	41.7%	16/0	27.0%	
Disease status, %							
Locally advanced	5	4.5%	2	4.2%	3	4.8%	>0.999
Metastatic	106	95.5%	46	95.8%	60	95.2%	
Location of metastases, %							
Liver	74	66.7%	38	79.2%	36	57.1%	0.015
Lung	18	16.2%	6	12.5%	12	19.0%	0.354
Lymph node	29	26.1%	11	22.9%	18	28.6%	0.502
Peritoneal dissemination	46	41.4%	25	52.1%	21	33.3%	0.047
Laboratory markers at diagnosis							
Platelets, 10^3^/µL, median (IQR)	226	(170–270)	221	(186–270)	230	(167–267)	0.812
NLR, median (IQR)	3.4	(2.3–5.6)	3.6	(2.4–5.7)	3.3	(2.3–5.5)	0.668
mGPS, %							0.036
0	70	63.1%	25	52.1%	45	71.4%	
1/2	24/17	36.9%	12/11	41.7%	12/6	27.0%	
CEA, ng/mL, median (IQR)	9.4	(3.4–48.6)	9.2	(2.8–50.2)	10.6	(3.6–40.8)	0.766
CA19-9, U/mL, median (IQR)	2193	(102–25,884)	2241	(74–17,897)	2193	(227–29,355)	0.833
Contrast-enhanced computed tomography findings at diagnosis							
Tumor size, mm (IQR)	46	(36–56)	53	(43–64)	42	(32–51)	<0.001
Splenic vein invasion, %	105	94.6%	46	95.8%	59	93.7%	0.697
Splenic vein occlusion, %	86	77.5%	41	85.4%	45	71.4%	0.080
Splenic artery invasion, %	92	82.9%	45	93.8%	47	74.6%	0.008
Splenic artery occlusion, %	26	23.4%	21	43.8%	5	7.9%	<0.001
Normal pancreatic tail remaining, %.	27	24.3%	1	2.1%	26	41.3%	<0.001
Dilated collateral veins, %	77	69.4%	36	75.0%	41	65.1%	0.261
Esophageal varices, %	2	1.8%	2	4.2%	0	0.0%	0.185
Gastric varices, %	54	48.6%	29	60.4%	25	39.7%	0.030
Splenic index (IQR)	236	(176–340)	253	(182–378)	232	(159–306)	0.051
Splenomegaly, %	6	5.4%	6	12.5%	0	0.0%	0.004
Gastrorenal shunt, %	6	5.4%	2	4.2%	4	6.3%	0.614

CA19-9: carbohydrate antigen 19-9, CEA: carcinoembryonic antigen, IQR: interquartile range, mGPS: modified Glasgow prognostic score, NLR: neutrophil-to-lymphocyte ratio.

**Table 2 cancers-15-05862-t002:** Imaging and endoscopic findings.

			Splenic Hilar Involvement				
	Total		Yes		No		
	(*n* = 111)		(*n* = 48)		(*n* = 63)		*p* Value
CT findings during follow-up							
Follow-up contrast-enhanced CT conducted, %	96	86.5%	42	87.5%	54	85.7%	0.785
Median follow-up period, months (IQR)	7.9	(4.6–14.1)	7.4	(3.1–10.4)	8.3	(4.7–16.7)	0.098
Esophageal varices, %	12	10.8%	7	14.6%	5	7.9%	0.264
Time to variceal formation, months (IQR) *	4.6	(2.6–7.4)	4.5	(3.5–7.4)	4.4	(2.3–6.9)	0.639
Maximum diameter, mm (IQR)	3.0	(2.6–4.0)	3.2	(2.9–3.6)	2.6	(2.6–3.6)	>0.999
Gastric varices, %	87	78.4%	42	87.5%	45	71.4%	0.042
Time to variceal formation, months (IQR) *	1.9	(1.8–3.5)	1.8	(1.6–3.4)	2.1	(1.8–3.3)	0.117
Maximum diameter, mm (IQR)	4.0	(3.4–5.1)	4.1	(3.4–5.4)	4.0	(3.4–4.9)	0.842
Sarin classification, %							0.120
IGV type 1	54	62.1%	26	61.9%	28	62.2%	
IGV type 2	4	4.6%	1	2.4%	3	6.7%	
IGV type 1 + isolated esophageal varices	3	3.4%	0	0.0%	3	6.7%	
IGV type 1 + IGV type 2	17	19.5%	8	19.0%	9	20.0%	
GOV type 1	1	1.1%	1	2.4%	0	0.0%	
GOV type 2	3	3.4%	2	4.8%	1	2.2%	
GOV type 1 + GOV type 2	5	5.7%	4	9.5%	1	2.2%	
Dilated collateral veins, %	81	73.0%	36	75.0%	45	71.4%	0.608
Main drainage route, %							
Left gastric vein	75	67.6%	32	76.2%	43	95.6%	0.020
Posterior gastric vein	12	10.8%	10	23.8%	2	4.4%	
Maximum splenic index, (IQR)	283	(212–422)	319	(220–476)	270	(180–393)	0.113
% total growth, (IQR)	20	(3–41)	22	(3–35)	19	(4–51)	0.909
% growth per month, (IQR)	2.2	(0.5–5.5)	2.7	(0.7–6.9)	1.8	(0.2–3.6)	0.209
Splenomegaly, %	18	16.2%	9	18.8%	9	14.3%	0.508
Endoscopic findings, %							
Emergency esophagogastroduodenoscopy performed	7	6.3%	3	6.3%	4	6.3%	>0.999
Upper gastrointestinal bleeding identified	6	5.4%	3	6.3%	3	4.8%	>0.999
Variceal rupture	1	0.9%	1	2.1%	0	0.0%	0.432
Tumor invasion	4	3.6%	2	4.2%	2	3.2%	>0.999
Gastric antral vascular ectasia	1	0.9%	0	0.0%	1	1.6%	>0.999
Portal vein thrombosis, %	2	1.8%	2	4.2%	0	0.0%	0.185
Biliary obstruction requiring drainage, %	11	9.9%	3	6.3%	8	12.7%	0.344
Duodenal obstruction requiring stenting, %	1	0.9%	0	0.0%	1	1.6%	>0.999
Colonic obstruction requiring stenting, %	7	6.3%	4	8.3%	3	4.8%	0.463
Splenic artery pseudoaneurysm rupture, %	3	2.7%	3	6.3%	0	0.0%	0.078

CT: computed tomography, GOV: gastroesophageal varices, IGV: isolated gastric varices, IQR: interquartile range. * Excludes patients who already had varices at the time of their cancer diagnosis. Denominators adjusted for missing data.

**Table 3 cancers-15-05862-t003:** Baseline characteristics, stratified by presence of gastric varices at diagnosis.

	Gastric Varices at Diagnosis				
	Yes		No		
	(*n* = 54)		(*n* = 57)		*p* Value
Age at diagnosis, years, median (IQR)	65	(55–71)	69	(62–74)	0.016
Male, %	34	63.0%	30	52.6%	0.271
Body mass index, median (IQR)	22.4	(19.6–24.8)	21.6	(19.3–23.3)	0.379
Performance status, %					
0	33	68.8%	42	66.7%	0.257
1/2	20/1	41.7%	15/0	27.0%	
Disease status, %					
Locally advanced	3	5.6%	2	3.5%	0.603
Metastatic	51	94.4%	55	96.5%	
Location of metastases, %					
Liver	37	68.5%	37	64.9%	0.687
Lung	12	22.2%	6	10.5%	0.095
Lymph node	10	18.5%	19	33.3%	0.076
Peritoneal dissemination	28	51.9%	18	31.6%	0.030
Other	7	13.0%	5	8.8%	0.336
Laboratory markers at diagnosis					
Platelets, 10^3^/µL, median (IQR)	216	(164–257)	24.1	(173–277)	0.145
NLR, median (IQR)	3.3	(2.3–6.1)	3.7	(2.4–5.4)	0.976
mGPS, %					
0	33	61.1%	37	64.9%	0.656
1/2	21	38.9%	20	35.1%	
CEA, ng/mL, median (IQR)	12.5	(3.5–49.2)	8.8	(3.3–38.9)	0.973
CA19-9, U/mL, median (IQR)	1455	(58–28,988)	3077	(353–25,269)	0.394
Contrast-enhanced computed tomography findings at diagnosis					
Tumor size, mm (IQR)	49	(41–59)	42	(31–51)	0.012
Splenic vein invasion, %	53	98.1%	52	91.2%	0.107
Splenic vein occlusion, %	49	90.7%	37	64.9%	0.001
Splenic artery invasion, %	50	92.6%	42	73.7%	0.008
Splenic artery occlusion, %	12	22.2%	14	24.6%	0.771
Splenic hilar invasion, %	29	53.7%	19	33.3%	0.030
Normal pancreatic tail remaining, %	8	14.8%	19	33.3%	0.023
Dilated collateral veins, %	51	94.4%	26	45.6%	<0.001
Esophageal varices, %	2	3.7%	0	0.0%	0.143
Splenic index (IQR)	270	(184–369)	208	(160–295)	0.015
Splenomegaly, %	5	9.3%	1	1.8%	0.081
Gastrorenal shunt, %	2	3.7%	4	7.0%	0.440

CA19-9: carbohydrate antigen 19-9, CEA: carcinoembryonic antigen, IQR: interquartile range, mGPS: modified Glasgow prognostic score, NLR: neutrophil-to-lymphocyte ratio.

**Table 4 cancers-15-05862-t004:** Imaging and endoscopic findings, stratified by presence of gastric varices at diagnosis.

	Gastric Varices at Diagnosis				
	Yes		No		
	(*n* = 54)		(*n* = 57)		*p* Value
CT findings during follow-up					
Follow-up contrast-enhanced CT conducted, %	46	85.2%	50	87.7%	0.696
Median follow-up period, months (IQR)	6.7	(4.1–10.5)	8.9	(4.7–16.1)	0.117
Esophageal varices, %	5	9.3%	7	12.3%	0.608
Time to variceal formation, months (IQR) *	4.5	(2.3–4.6)	4.6	(3.4–7.8)	0.149
Maximum diameter, mm (IQR)	3.8	(3.3–4.6)	2.6	(2.5–3.3)	0.381
Gastric varices, %	54	100.0%	33	57.9%	<0.001
Time to variceal formation, months (IQR) *	-	-	1.9	(1.8–3.5)	-
Maximum diameter, mm (IQR)	4.1	(3.4–5.3)	3.8	(3.4–4.2)	0.126
Sarin classification, %					0.725
IGV type 1	31	57.4%	23	69.7%	
IGV type 2	3	5.6%	1	3.0%	
IGV type 1 + isolated esophageal varices	0	0.0%	3	9.1%	
IGV type 1 + IGV type 2	15	27.8%	2	6.1%	
GOV type 1	1	1.9%	0	0.0%	
GOV type 2	0	0.0%	3	9.1%	
GOV type 1 + GOV type 2	4	7.4%	1	3.0%	
Dilated collateral veins, %	46	85.2%	35	61.4%	<0.001
Main drainage from left gastric vein, %	36	78.3%	33	94.3%	<0.001
Main drainage from posterior gastric vein, %	10	21.7%	2	5.7%	<0.001
Maximum splenic index, (IQR)	332	(250–476)	257	(190–379)	0.011
% total growth, (IQR)	19	(4–47)	20	(4–39)	0.953
% growth per month, (IQR)	2.2	(0.6–5.6)	2.1	(0.3–5.3)	0.741
Splenomegaly, %	11	20.4%	7	12.3%	0.185
Endoscopic findings, %					
Emergency esophagogastroduodenoscopy performed	3	5.6%	4	7.0%	0.751
Upper gastrointestinal bleeding identified	2	3.7%	4	7.0%	0.440
Variceal rupture	1	1.9%	0	0.0%	0.486
Tumor invasion	1	1.9%	3	5.3%	0.335
Gastric antral vascular ectasia	0	0.0%	1	1.8%	>0.999
Portal vein thrombosis, %	2	3.7%	0	0.0%	0.143
Biliary obstruction requiring drainage, %	2	3.7%	9	15.8%	0.033
Duodenal obstruction requiring stenting, %	1	1.9%	0	0.0%	0.486
Colonic obstruction requiring stenting, %	4	7.4%	3	5.3%	0.642
Splenic artery aneurysm rupture, %	1	1.9%	2	3.5%	0.591

CT: computed tomography, GOV: gastroesophageal varices, IGV: isolated gastric varices, IQR: interquartile range. * Excludes patients who already had varices at the time of their cancer diagnosis. Denominators adjusted for missing data.

**Table 5 cancers-15-05862-t005:** Chemotherapy regimens and responses to treatment.

	Splenic Hilar Involvement				
	Yes		No		
	(*n* = 48)		(*n* = 63)		*p* Value
Chemotherapy regimen, %					
Gemcitabine + nab-paclitaxel	34	70.8%	54	85.7%	0.055
Modified FOLFIRINOX	10	20.8%	6	9.5%	
Gemcitabine monotherapy	2	4.2%	2	3.2%	
S-1 monotherapy	2	4.2%	0	0.0%	
Radiotherapy, %	0	0.0%	1	1.6%	
Best response, %					
Complete response	0	0.0%	0	0.0%	-
Partial response	10	20.8%	15	23.8%	0.446
Stable disease	18	37.5%	34	54.0%	0.063
Progressive disease	18	37.5%	9	14.3%	0.005
Not evaluable	2	4.2%	5	7.9%	
Overall response rate, % (a)	10	20.8%	15	23.8%	0.625
Disease control rate, % (a)	28	58.3%	49	77.8%	0.006

(a) Denominators adjusted to exclude patients whose best responses were not evaluable.

**Table 6 cancers-15-05862-t006:** Factors affecting overall survival.

	Univariate			Multivariate		
	Hazard Ratio	95% CI	*p*-Value	Hazard Ratio	95% CI	*p*-Value
Old age (70 years or older)	0.78	0.51–1.16	0.211			
Male sex	1.04	0.73–1.54	0.839			
Poor performance status (1 or 2)	2.21	1.46–3.34	<0.001	1.86	1.19–2.89	0.006
Tumor size (>50 mm)	1.70	1.14–2.55	0.010			
Metastatic (vs. locally advanced)	1.29	0.56–2.95	0.554			
Gemcitabine + nab-paclitaxel (vs. other regimens)	1.31	0.82–2.10	0.264			
Hepatic metastasis	2.29	1.48–3.53	<0.001	1.93	1.22–3.03	0.005
Lung metastasis	0.62	0.36–1.06	0.083			
Lymph node metastasis	1.17	0.75–1.81	0.495			
Peritoneal dissemination	1.10	0.75–1.63	0.629			
NLR > 4	2.06	1.39–3.05	<0.001	1.47	0.94–2.30	0.089
Platelets < 130,000/µL	0.34	0.12–0.92	0.038			
mGPS (1 or 2)	2.68	1.78–4.05	<0.001	1.96	1.25–3.08	0.003
CEA (>5 ng/mL)	1.49	0.97–2.20	0.070			
CA19-9 (>500 U/mL)	1.59	1.05–2.41	0.027			
Obesity (BMI > 25)	1.57	0.96–2.56	0.070			
Splenic vein occlusion	1.25	0.79–1.99	0.349			
Splenic hilar involvement	1.79	1.20–2.65	0.004	1.65	1.08–2.52	0.020
Dilated collateral veins	0.93	0.61–1.41	0.738			
Splenic artery occlusion	1.00	0.63–1.59	0.991			
Gastric varices at diagnosis	1.02	0.69–1.51	0.908			
Splenomegaly	1.25	0.55–2.86	0.600			

BMI: body mass index, CA19-9: carbohydrate antigen 19-9, CEA: carcinoembryonic antigen, CI: confidence interval, mGPS: modified Glasgow prognostic score, NLR: neutrophil-to-lymphocyte ratio.

**Table 7 cancers-15-05862-t007:** Factors affecting progression-free survival.

	Univariate			Multivariate		
	Hazard Ratio	95% CI	*p*-Value	Hazard Ratio	95% CI	*p*-Value
Old age (70 years or older)	0.63	0.41–0.96	0.031			
Male sex	0.85	0.57–1.29	0.447			
Poor performance status (1 or 2)	2.07	1.34–3.19	<0.001	1.86	1.19–2.91	0.006
Tumor size (>50 mm)	1.73	1.13–2.64	0.011	1.80	1.15–2.80	0.009
Metastatic (vs. locally advanced)	0.77	0.28–2.11	0.615			
Gemcitabine + nab-paclitaxel (vs. other regimens)	1.07	0.66–1.75	0.783			
Hepatic metastasis	1.49	0.96–2.30	0.075			
Lung metastasis	0.56	0.31–1.02	0.057			
Lymph node metastasis	1.16	0.73–1.84	0.527			
Peritoneal dissemination	0.90	0.59–1.37	0.605			
NLR > 4	1.62	1.06–2.47	0.025			
Platelets < 130,000/µL	0.29	0.11–0.81	0.018	0.35	0.12–0.998	0.0499
mGPS (1 or 2)	2.39	1.56–3.67	<0.001	2.13	1.38–3.29	0.001
CEA (>5 ng/mL)	1.31	0.85–2.02	0.224			
CA19-9 (>500 U/mL)	1.50	0.97–2.33	0.069			
Obesity (BMI > 25)	0.98	0.57–1.68	0.933			
Splenic vein occlusion	1.60	0.97–2.63	0.066			
Splenic hilar involvement	1.67	1.11–2.51	0.015	1.49	0.98–2.28	0.065
Dilated collateral veins	1.29	0.81–2.03	0.276			
Splenic artery occlusion	1.20	0.75–1.94	0.446			
Gastric varices at diagnosis	0.94	0.62–1.42	0.775			
Splenomegaly	1.27	0.55–2.92	0.580			

BMI: body mass index, CA19-9: carbohydrate antigen 19-9, CEA: carcinoembryonic antigen, CI: confidence interval, mGPS: modified Glasgow prognostic score, NLR: neutrophil-to-lymphocyte ratio.

**Table 8 cancers-15-05862-t008:** Response to chemotherapy, stratified by the presence or absence of gastric varices.

	Gastric Varices at Diagnosis				
	Yes		No		
	(*n* = 54)		(*n* = 57)		*p* Value
Chemotherapy regimen, *n* (%)					
Gemcitabine + nab-paclitaxel	44	81.5%	44	77.2%	0.374
Modified FOLFIRINOX	7	13.0%	9	15.8%	
Gemcitabine monotherapy	2	3.7%	2	3.5%	
S-1 monotherapy	1	1.9%	1	1.8%	
Radiotherapy, *n* (%)	0	0.0%	1	1.8%	>0.999
Best response, *n* (%)					
Complete response	0	0.0%	0	0.0%	-
Partial response	11	20.4%	14	24.6%	0.382
Stable disease	24	44.4%	28	49.1%	0.381
Progressive disease	15	27.8%	12	21.1%	0.273
Not evaluable	4	7.4%	3	5.3%	
Overall response rate, n (%) (a)	11	22.0%	14	25.9%	0.640
Disease control rate, n (%) (a)	35	70.0%	42	77.8%	0.366

(a) Denominators adjusted to exclude patients whose best responses were not evaluable.

## Data Availability

The data used in this study are contained within the tables presented herein. Additional data pertaining to this study are available upon reasonable request from the corresponding author.

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
