# Peer review of "Splenic Hilar Involvement and Sinistral Portal Hypertension in Unresectable Pancreatic Tail Cancer"

_cancers, 2023, doi:10.3390/cancers15245862_

Round 1

Reviewer 1 Report

Comments and Suggestions for Authors

The authors present results from a single center retrospective study evaluating the prognostic impact of splenic hilar involvement in a limited number of patients with pancreatic tail cancer (n=111). Literature regarding this topic is scarce, so the study adds scientific knowledge to this topic. The work is of particular interest to readers who are interested in treatment/ prognosis of advanced pancreatic cancer. However, I have some comments:

1. The authors focused on the presence and prognostic relevance of sinistral portal hypertension- how was SPH defined? Only by presence of GV in Imaging? Please specify.

2. SHI patients showed lower disease control rates in patients with SHI. It is also said that these tumours are probably more aggressive and are therefore associated with a poorer prognosis. A comparison of these two groups is difficult and one should be careful with the conclusions, because the groups were not randomised and the type of chemotherapy was specified, but not the cycles applied.

3. In univariate Analysis, metastatic vs. locally advanced disease did affect survival although it is generally known that metastasised patients have a poorer survival rate - do the authors have an explanation for this? Possibly the low number of locally advanced stages in the cohort?

 4. The authors show 8 Tables and 4 Figures- In order to improve clarity, the transfer of some tables/figures to the supplement should be considered (e.g. Table 8 and/or Figure 4).

 5. SHI is associated with poorer survival, but does not seem  to be due to bleeding of GV, because the number of GI bleed was relatively low, despite high rates of GV in both groups. Is there a possibility that this low number was due to missing data? Do the authors have implications fort he clinical practice for these patients: e.g. regular screening endoscopies, administration of non-selective betablockers as bleeding prophylaxis?

Author Response

We would like to thank the reviewers for their careful evaluation and helpful comments concerning our manuscript. We have revised our manuscript based on the comments received, as detailed below.

Reviewer 1:

The authors present results from a single center retrospective study evaluating the prognostic impact of splenic hilar involvement in a limited number of patients with pancreatic tail cancer (n=111). Literature regarding this topic is scarce, so the study adds scientific knowledge to this topic. The work is of particular interest to readers who are interested in treatment/ prognosis of advanced pancreatic cancer. However, I have some comments:

  1. The authors focused on the presence and prognostic relevance of sinistral portal hypertension- how was SPH defined? Only by presence of GV in Imaging? Please specify.

Response:

Thank you for your comment. In this study, we could not analyze the presence or absence of sinistral portal hypertension (SPH) per se, because we did not measure the portal-splenic venous gradient. The presence of gastric varices on images was used as a surrogate marker for SPH. To clarify this, we have added the following to our manuscript (line 103):

Varices were classified according to Sarin et al.’s classification [26] and were used as a surrogate marker for SPH in this study.

  1. SHI patients showed lower disease control rates in patients with SHI. It is also said that these tumours are probably more aggressive and are therefore associated with a poorer prognosis. A comparison of these two groups is difficult and one should be careful with the conclusions, because the groups were not randomised and the type of chemotherapy was specified, but not the cycles applied.

Response:

We agree that this was not a randomized study and that the number of cycles was not specified. With respect to number of cycles, we did uniformly provide treatment until disease progression, patient refusal, intolerable toxicity, inability to continue treatment for other reasons, or conversion surgery. This was the case for all agents, including cisplatin (it was not necessarily stopped after 8 cycles). To clarify this, we added the following to our manuscript (line 123):

Treatment was continued until disease progression, patient refusal, intolerable toxicity, inability to continue treatment for other reasons, or conversion surgery.

We also agree that comparison of the SHI and non-SHI groups is difficult and that conclusions should be drawn with caution. We have added the following sentence to our limitations section (line 327):

Reports on SHI are scarce and there is no consensus to date on the definition of SHI, mak-ing interpretations of our results difficult.

  1. In univariate Analysis, metastatic vs. locally advanced disease did affect survival although it is generally known that metastasised patients have a poorer survival rate - do the authors have an explanation for this? Possibly the low number of locally advanced stages in the cohort?

Response:

As the reviewer correctly points out, we believe the lack of significance of this factor is due to the low number of locally advanced patients. Locally advanced pancreatic tail cancer is a very rare entity, as we state in line 50: “The pancreatic head and body are anatomically close to key arteries such as the celiac axis, superior mesenteric artery, and their respective branches, as well as the splenoportal con-fluence (SPC). On the other hand, PDAC of the pancreatic tail (pancreatic tail cancer: PTC) initially only invades the splenic vein and/or artery, making them less likely to be border-line resectable [12] or unresectable due to locally-advanced disease.” We felt that this does not need to be reiterated in our discussion, as Table 1 makes it clear that we have very few locally-advanced cases. The large 95% confidence intervals in Tables 4 and 5 also provide clues that one subgroup has a small sample size.

  1. The authors show 8 Tables and 4 Figures- In order to improve clarity, the transfer of some tables/figures to the supplement should be considered (e.g. Table 8 and/or Figure 4).

Response:

Thank you for this suggestion. We agree that we have a large number of tables and figures. However, Cancers does not have a limitation on the number of table and figures. It has been our experience that readers find it a hassle to open supplements and do not read them, so we would like to avoid them if possible. Because this comment is not related to the format and not the content of the manuscript, we will follow the advice of the editorial team as needed. We do appreciate the kind suggestion.

  1. SHI is associated with poorer survival, but does not seem to be due to bleeding of GV, because the number of GI bleed was relatively low, despite high rates of GV in both groups. Is there a possibility that this low number was due to missing data? Do the authors have implications for the clinical practice for these patients: e.g. regular screening endoscopies, administration of non-selective betablockers as bleeding prophylaxis?

Response:

The reviewer correctly points out that poorer survival is not the result of bleeding GVs. We do not believe this is due to missing data, as very few patients were lost to follow up. This is most likely because unresectable pancreatic cancer do not live long enough to experience bleeding events. Also, as stated in our discussion, gastric varices are known to have a lower risk of rupture compared to esophageal varices, in part due to the thicker gastric walls.

We agree that it is important to consider implications for clinical practice. We do not have data on the use of beta blockers in our sample, and therefore cannot make recommendations on their use. However, given the low incidence of gastric variceal rupture, we do not believe that regular EGD screening would be cost-effective in this population. It would also add to the burden of these patients, who either make weekly to biweekly hospital visits for chemotherapy or are in very poor condition in the first place. With respect to EGD screening, we added the following to our discussion (line 303):

In our study, GV rupture occurred only in 0.9% of unresectable PTC cases and in 2.1% of SHI cases. While endoscopic variceal treatment is generally not very effective in SPH due to a myriad of dilated vessels involved [31], our sole patient with GV rupture survived 7.2 months after the UGIB event. We speculate that most unresectable PTC patients do not live long enough to experience variceal rupture. Given the costs and patient burden involved, we believe it is premature to recommend routine EGD screening in all unresectable PTC patients, including those with SHI.

Reviewer 2 Report

Comments and Suggestions for Authors

This manuscript describes the significance of splenic hilar involvement as a prognostic factor in unresectable pancreatic tail cancer. Although the content is interesting, I have a few comments to the author.

Comment 1. I think the most important point in this research is to clearly define and distinguish between the presence and absence of SHI. The Fig 1 shows a CT image of with/without SHI that can be clearly distinguished, but was there actually any confusion about the diagnosis of with/without SHI?

Comment 2. In Fig 3, it is easier to understand which of the red line and blue line is the SHI group and which is the non-SHI group in the figure legend. Similarly, in Fig 4, it would be easier to understand if the figure legend indicates which of the red line and blue line is the gastric varices group and which is the no gastric varices group.

Comment 3. Figure 3 shows that when SHI positive, both OS and PSF are poor, but the median difference in OS and PSF between with or without SHI is about 2 months. What is the clinical significance of with/without SHI? I think it is necessary to consider whether there are any measures to improve the prognosis in with SHI cases.

Author Response

We would like to thank the reviewers for their careful evaluation and helpful comments concerning our manuscript. We have revised our manuscript based on the comments received, as detailed below.

Reviewer 2:

This manuscript describes the significance of splenic hilar involvement as a prognostic factor in unresectable pancreatic tail cancer. Although the content is interesting, I have a few comments to the author.

Comment 1. I think the most important point in this research is to clearly define and distinguish between the presence and absence of SHI. The Fig 1 shows a CT image of with/without SHI that can be clearly distinguished, but was there actually any confusion about the diagnosis of with/without SHI?

Response:

As the reviewer correctly points out, being able to determine whether or not SHI is present is central to the interpretation of this study. As the reviewer has kindly confirmed, Fig 1 shows very typical examples. However, we believe we have provided an adequately clear definition of SHI, including the fact that involvement of the spleen itself was not necessary. Because SHI required “stricture or occlusion” of the splenic hilar vasculature, abutment of the tumor with the splenic hilum was not considered adequate to meet the definition of SHI. In our analyses of diagnostic imaging, we did not experience trouble when differentiating between SHI and non-SHI based on our definition.

Comment 2. In Fig 3, it is easier to understand which of the red line and blue line is the SHI group and which is the non-SHI group in the figure legend. Similarly, in Fig 4, it would be easier to understand if the figure legend indicates which of the red line and blue line is the gastric varices group and which is the no gastric varices group.

Response:

Thank you for pointing this out. We agree, and have made the suggested changes.

Comment 3. Figure 3 shows that when SHI positive, both OS and PSF are poor, but the median difference in OS and PSF between with or without SHI is about 2 months. What is the clinical significance of with/without SHI? I think it is necessary to consider whether there are any measures to improve the prognosis in with SHI cases.

Response:

We agree that the differences in both median OS and median PFS between the SHI and non-SHI groups is about 2 months. In the realm of unresectable pancreatic cancer, we believe this is a clinically meaningful difference. For example, nab-paclitaxel plus gemcitabine only improved OS by 1.8 months and PFS by 1.8 months versus gemcitabine alone in the landmark phase 3 study (N Engl J Med. 2013 Oct 31; 369(18): 1691–1703.).

We also agree that measures to improve prognosis in SHI cases are desirable. However, our sample is mostly composed of metastatic pancreatic cancer, with a poor prognosis and very limited treatment options (limited to chemotherapy). We also found that a potential complication of SHI, namely bleeding from GV, was rare and did not significantly affect OS or PFS. Thus, preventing GV rupture does not seem to prolong OS, and we currently have no suggestions on how to improve OS in this population. We believe that this study does contribute to educating physicians of the implications of SHI and that treatment of asymptomatic GVs is not necessarily warranted.

We added the following on discussion on the need for EGD screening to our discussion (line 303):

In our study, GV rupture occurred only in 0.9% of unresectable PTC cases and in 2.1% of SHI cases. While endoscopic variceal treatment is generally not very effective in SPH due to a myriad of dilated vessels involved [31], our sole patient with GV rupture survived 7.2 months after the UGIB event. We speculate that most unresectable PTC patients do not live long enough to experience variceal rupture. Given the costs and patient burden involved, we believe it is premature to recommend routine EGD screening in all unresectable PTC patients, including those with SHI.

Reviewer 3 Report

Comments and Suggestions for Authors

Manuscript ID: cancers-2741919

Title: Splenic hilar involvement and sinistral portal hypertension in unresectable pancreatic tail cancer

Authors: Okamoto et al. 

The authors repot that splenic hilar involvement and sinistral portal hypertension is associated with worse outcome in patients with unresectable pancreatic tail cancer. This observation is not surprising, because logic suggests that higher degree of cancer development, and involvement of adjacent organs and blood vessels is associated with a poor prognosis for the patients.

On the other hand, it should be noted that the assumption presented above require verification and the study by Okamoto et al. provides hard evidence confirming the relationship between the involvement of the splenic hilum and sinistral portal hypertension and the severity of unresectable pancreatic tail cancer. The manuscript is well written and based on data collected from the patients with unresectable PDAC of the pancreatic tail who underwent chemotherapy at Department of Hepato-Biliary-Pancreatic Medicine, Cancer Institute Hospital of Japanese Foundation for Cancer Research in Tokyo between January 1, 2016, and December 68 31, 2020. The authors used bibliography reflecting the current stage of knowledge in the topic of research presented. According to the reviewer’s opinion, the manuscript is almost ready for publication. There are only minimal shortcomings that should be corrected.

  1. Too many abbreviations make the manuscript difficult to read. At least in conclusions in the abstract and text, abbreviations should be eliminated.
  2. Table 1 and Table 3. What does “Platelets, 10^3/µL” mean? Does “10^3” mean 103? Does this mean that the patients had about 22,000 platelets per 1 µL of blood?
  3. Figure 3. In Figures 3a and 3b or in the figure legend, the authors should write that the red curve represents effects in patients with SHI, whereas the blue curve represents the effects in patients without SHI.

Author Response

We would like to thank the reviewers for their careful evaluation and helpful comments concerning our manuscript. We have revised our manuscript based on the comments received, as detailed below.

Reviewer 3:

The authors repot that splenic hilar involvement and sinistral portal hypertension is associated with worse outcome in patients with unresectable pancreatic tail cancer. This observation is not surprising, because logic suggests that higher degree of cancer development, and involvement of adjacent organs and blood vessels is associated with a poor prognosis for the patients.

On the other hand, it should be noted that the assumption presented above require verification and the study by Okamoto et al. provides hard evidence confirming the relationship between the involvement of the splenic hilum and sinistral portal hypertension and the severity of unresectable pancreatic tail cancer. The manuscript is well written and based on data collected from the patients with unresectable PDAC of the pancreatic tail who underwent chemotherapy at Department of Hepato-Biliary-Pancreatic Medicine, Cancer Institute Hospital of Japanese Foundation for Cancer Research in Tokyo between January 1, 2016, and December 68 31, 2020. The authors used bibliography reflecting the current stage of knowledge in the topic of research presented. According to the reviewer’s opinion, the manuscript is almost ready for publication. There are only minimal shortcomings that should be corrected.

Response:

Thank you for your supportive comments.

  1. Too many abbreviations make the manuscript difficult to read. At least in conclusions in the abstract and text, abbreviations should be eliminated.

Response:

Thank you for your suggestion. We have eliminated abbreviations in the conclusion sections of the abstract and text. Because of the word limit of 200 words for the abstract, we revised the abstract slightly to allow for this change.

  1. Table 1 and Table 3. What does “Platelets, 10^3/µL” mean? Does “10^3” mean 103? Does this mean that the patients had about 22,000 platelets per 1 µL of blood?

Response:

Thank you for your detailed review. Yes, 10^3 means 10 to the third power, or 1000. However, our figures were off by a factor of ten. Our patients had about 220,000 platelets per 1 µL of blood. This error occurred because platelets are traditionally shown as 10^4/µL in Japan. We have corrected this error in the tables and in the manuscript.

  1. Figure 3. In Figures 3a and 3b or in the figure legend, the authors should write that the red curve represents effects in patients with SHI, whereas the blue curve represents the effects in patients without SHI.

Response:

Thank you for pointing this out. We agree, and have made the suggested changes. We also made similar changes to the figure legend to Figure 4.

Round 2

Reviewer 2 Report

Comments and Suggestions for Authors

Thank you for answering to my comments.